# Autonomous artificial intelligence increases screening and follow-up for diabetic retinopathy in youth: the ACCESS randomized control trial

Risa M. Wolf [1] ✉, Roomasa Channa[2], T. Y. Alvin Liu[3], Anum Zehra[1], Lee Bromberger[1], Dhruva Patel[1], Ajaykarthik Ananthakrishnan [1], Elizabeth A. Brown [1], Laura Prichett [4], Harold P. Lehmann[5] & Michael D. Abramoff [6,7,8,9,10]

Diabetic retinopathy can be prevented with screening and early detection. We hypothesized that autonomous artificial intelligence (AI) diabetic eye exams at the point-of-care would increase diabetic eye exam completion rates in a racially and ethnically diverse youth population. AI for Children's diabetiC Eye ExamS (NCT05131451) is a parallel randomized controlled trial that randomized youth (ages 8-21 years) with type 1 and type 2 diabetes to intervention (autonomous artificial intelligence diabetic eye exam at the point of care), or control (scripted eye care provider referral and education) in an academic pediatric diabetes center. The primary outcome was diabetic eye exam completion rate within 6 months. The secondary outcome was the proportion of participants who completed follow-through with an eye care provider if deemed appropriate. Diabetic eye exam completion rate was significantly higher (100%, 95%CI: 95.5%, 100%) in the intervention group ($n = 81$) than the control group ($n = 83$) (22%, 95%CI: 14.2%, 32.4%)(p < 0.001). In the intervention arm, 25/81 participants had an abnormal result, of whom 64% (16/25) completed follow-through with an eye care provider, compared to 22% in the control arm (p < 0.001). Autonomous AI increases diabetic eye exam completion rates in youth with diabetes.

Diabetic eye disease (DED) is a complication of diabetes that is the primary cause of blindness in working-age adults in the U.S.[1,2]. Early detection ('screening') and treatment can frequently prevent progression, but the majority of the 34 million people with diabetes in the US have a DED screening care gap, due to a lack of access, and education around the need for a diabetic eye exam[3,4]. This care gap is a major source of health disparity, with racial and ethnic minorities, and under-resourced communities, having

[1]Department of Pediatrics, Division of Endocrinology, Johns Hopkins School of Medicine, Baltimore, MD, USA. [2]Department of Ophthalmology and Visual Sciences, University of Wisconsin, Madison, WI, USA. [3]Wilmer Eye Institute at the Johns Hopkins School of Medicine, Baltimore, MD, USA. [4]Johns Hopkins School of Medicine Biostatistics, Epidemiology and Data Management (BEAD) Core, Baltimore, MD, USA. [5]Section on Biomedical Informatics and Data Science, Johns Hopkins University, Baltimore, MD, USA. [6]Department of Ophthalmology and Visual Sciences, The University of Iowa, Iowa City, IA, USA. [7]Digital Diagnostics Inc, Coralville, IA, USA. [8]Iowa City VA Medical Center, Iowa City, IA, USA. [9]Department of Biomedical Engineering, The University of Iowa, Iowa City, IA, USA. [10]Department of Electrical and Computer Engineering, The University of Iowa, Iowa City, IA, USA. ✉e-mail: RWolf@jhu.edu

**Fig. 1 | Flow diagram of patient enrollment and randomization in the ACCESS study (Based on CONSORT guideline 2010 flow diagram).**

worse outcomes, and a disproportionately higher prevalence of DED[2,5–9].

While DED prevalence is lower in youth (defined as those aged <21 years) with diabetes, it affects approximately 4–9% of youth with type 1 diabetes (T1D) and 4-15% of youth with type 2 diabetes (T2D)[10–14]. The risk for DED increases with the duration of diabetes in T1D, and recent data from the TODAY2 follow-up study demonstrated a diabetic retinopathy prevalence rate of 49% at a mean diabetes duration of 12 years in youth onset T2D[15]. The American Diabetes Association (ADA) and American Academy of Ophthalmology (AAO) recommend DED screening in T1D within 3–5 years of diagnosis and age greater than 11 years, and in T2D at the time of diagnosis in youth[16]. However, only 35-72% of diabetic youth undergo recommended screening exams, with even higher care gap rates in minority and lower socioeconomic background youth[10,17]. Commonly reported barriers to screening include miscommunication regarding the need for a diabetic eye exam, time for an additional doctor's visit, and transportation barriers[17,18].

While the introduction of telemedicine over the last two decades has improved screening and facilitated early detection of diabetic eye disease[19–23], the development of diagnostic autonomous artificial intelligence (AI) systems for diagnosing DED has ushered in the next chapter of DED screening[24–29]. Autonomous AI systems require a camera operator to obtain point-of-care fundus images, which are then interpreted by an AI algorithm to provide a diagnosis without human oversight. The regulatory approval of the first diagnostic autonomous AI system for diabetic eye exams was based on a pivotal trial against a prognostic standard, i.e., patient outcome[24], showing its safety, efficacy, and lack of racial and ethnic bias for diagnosing DED in adults with diabetes with 87% sensitivity and 91% specificity[24]. In prior studies of youth with diabetes, we demonstrated that the diagnosability of

autonomous AI in youth was 97.5%, with 85.7% sensitivity and 79.3% specificity in detecting DED, with no difference in diagnosability across demographic groups[30]. We have also shown that this system can be implemented in a multidisciplinary diabetes clinic and has the potential to increase DED screening rates in underserved youth, while also being cost-savings to patients and caregivers[30,31].

While diagnostic accuracy has been a focus of study of diagnostic AI systems[25,30,32], the effectiveness of autonomous AI to increase adherence and follow-up compared to traditional referral has not been evaluated in a rigorously designed randomized trial. We hypothesized that autonomous AI closes the diabetic eye exam care gap, and increases follow-up, compared to traditional eye care provider(ECP) referral in youth. To test this hypothesis, we designed ACCESS (AI for Childrens' diabetiC Eye examS Study), a pre-registered, rigorously designed randomized control trial (RCT), to measure diabetic eye exam completion rates in a racially and ethnically diverse cohort of youth with T1D and T2D.

## Results
### Participant flow
One-hundred seventy candidates were determined to be eligible, and 164 participants completed informed consent and were randomized, 81 to the intervention and 83 to the control arm. The final allocation was not equal between the 2 groups due to recruitment completion in the middle of the permutated blocks. Six patients who were approached declined participation as described in the flowchart in Fig. 1.

### Baseline patient characteristics
Baseline characteristics were similar in the two groups (Table 1): mean age 15.2 years (SD 2.8), 58% female, 35% Black, 6% Hispanic, 47%

**Table 1 | ACCESS patient characteristics by randomization group**

| Factor | All participants | Standard of care | Autonomous AI |
|---|---|---|---|
| N | 164 | 83 | 81 |
| Age, mean (SD) | 15.2 (2.8) | 15.1 (2.8) | 15.3 (2.8) |
| **Race** | | | |
| Asian | 10 (6.1%) | 4 (4.8%) | 6 (7.4%) |
| NH Black | 58 (35.4%) | 32 (38.6%) | 26 (32.1%) |
| Hispanic | 10 (6.1%) | 5 (6.0%) | 5 (6.2%) |
| NH White | 86 (52.4%) | 42 (50.6%) | 44 (54.3%) |
| Male sex | 68 (41.5%) | 36 (43.4%) | 32 (39.5%) |
| **Household income** | | | |
| Less than $25,000 | 25 (15.2%) | 14 (16.9%) | 11 (13.6%) |
| $25,000–$49,999 | 31 (18.9%) | 16 (19.3%) | 15 (18.5%) |
| $50,000–$74,999 | 28 (17.1%) | 15 (18.1%) | 13 (16.0%) |
| $75,000–$99,999 | 16 (9.8%) | 5 (6.0%) | 11 (13.6%) |
| More than $100,000 | 48 (29.3%) | 27 (32.5%) | 21 (25.9%) |
| Choose not to answer/refused | 16 (9.8%) | 6 (7.2%) | 10 (12.3%) |
| **Highest education** | | | |
| Less than 12 years of high school | 5 (3.0%) | 3 (3.6%) | 2 (2.5%) |
| High school/GED | 57 (34.8%) | 27 (32.5%) | 30 (37.0%) |
| Associate's degree | 19 (11.6%) | 10 (12.0%) | 9 (11.1%) |
| Undergraduate degree | 25 (15.2%) | 13 (15.7%) | 12 (14.8%) |
| Post-graduate degree | 51 (31.1%) | 27 (32.5%) | 24 (29.6%) |
| Unknown | 7 (4.3%) | 3 (3.6%) | 4 (4.9%) |
| **Medicaid insurance** | 77 (47.0%) | 40 (48.2%) | 37 (45.7%) |
| Type 1 diabetes | 119 (72.6%) | 60 (72.3%) | 59 (72.8%) |
| Type 2 diabetes | 45 (27.4%) | 23 (27.7%) | 22 (27.2%) |
| Duration of diabetes (years), median (IQR) | 5.8 (3.2, 8.7) | 6.2 (2.9, 9.4) | 5.3 (3.4, 7.9) |
| HbA1c value at study visit (%), mean (SD) | 8.6 (2.3) | 8.5 (2.2) | 8.7 (2.3) |
| Continuous glucose monitor use | 125 (76.2%) | 63 (75.9%) | 62 (76.5%) |
| Has ever had a prior diabetic eye exam | 129 (78.7%) | 63 (75.9%) | 66 (81.5%) |

**Table 2 | Univariate analysis of participant characteristics based on prior diabetic eye exam completion**

| Variable | Baseline prior eye exam N = 164 | | |
| | No previous DED screening | Previous DED screening | p-value[a] |
|---|---|---|---|
| N | 35 | 129 | |
| Age, mean (SD) | 14.1 (2.4) | 15.5 (2.8) | 0.005 |
| **Race** | | | |
| Asian | 2 (5.7%) | 8 (6.2%) | 0.91 |
| NH Black | 22 (62.9%) | 36 (27.9%) | <0.001 |
| Hispanic | 5 (14.3%) | 5 (3.9%) | 0.022 |
| NH White | 6 (17.1%) | 80 (62.0%) | <0.001 |
| Male sex | 13 (37.1%) | 55 (42.6%) | 0.56 |
| **Household income** | | | 0.005 |
| <$50,000 per year | 20 (57.1%) | 36 (27.9%) | |
| >=$50,000 per year | 13 (37.1%) | 79 (61.2%) | |
| Unknown | 2 (5.7%) | 14 (10.9%) | |
| **Highest education** | | | 0.020 |
| High school or less | 16 (45.7%) | 46 (35.7%) | |
| More than high school | 15 (42.9%) | 80 (62.0%) | |
| Unknown | 4 (11.4%) | 3 (2.3%) | |
| **Medicaid insurance** | 29 (82.9%) | 48 (37.2%) | <0.001 |
| Type 1 Diabetes | 13 (37.1%) | 106 (82.2%) | <0.001 |
| Type 2 Diabetes | 22 (62.9%) | 23 (17.8%) | <0.001 |
| Duration of diabetes (years), median (IQR) | 1.4 (0.6, 4.4) | 7.3 (4.2, 9.2) | <0.001 |
| HbA1c value at this visit, mean (SD) | 8.2 (2.9) | 8.7 (2.1) | 0.25 |
| Continuous glucose monitor use | 15 (42.9%) | 110 (85.3%) | <0.001 |

[a]p-values were calculated using Chi-Squared tests for categorical variables, Wilcoxon rank-sum test for duration of diabetes, and Student's t-tests for all other continuous variables. All statistical tests are two-sided.

Medicaid insurance, 34% with household income <$50,000, 38% with parental education of up to high school completion, 73% had T1D, the median duration of diabetes was 5.8 years, and mean HbA1c was 8.6%.

Regarding disparities in ever having a prior diabetic eye exam, 79% of participants reported having had such an exam at some time in the past (but per inclusion criteria, not in the prior 6 months) and 35/164 (21%) did not (i.e., had a care gap). Those without a prior eye exam were more likely to be Black ($p = $ <0.001), Hispanic ($p = 0.02$), have household income of less than $50,000 ($p = 0.005$), Medicaid insurance ($p < 0.001$), parental education of up to high school completion ($p = 0.02$), and a shorter duration of diabetes ($p < 0.001$) in univariate analysis. (Table 2) When controlling for these factors in multivariate analysis, only the duration of diabetes remained significantly associated with prior eye exam completion (OR 1.36; 95% CI: 1.0–1.8, $p = 0.04$), suggesting that racial, ethnic, and SES disparities were driven by differences in duration of diabetes in the respective populations. There were differences in diabetes type, race, parental education, and diabetes management characteristics between the two recruitment sites, but glycemic control (measured by hemoglobin A1c) was similar across sites. ([Supplementary Note 8 and Table S1]).

## Primary outcome

As shown in Table 3, in the intervention arm, 81/81 (100%) participants completed their diabetic eye exams, so the primary care gap closure rate was 100% (95%CI: 96%, 100%). All images were diagnostic for the AI system (output was either "DED present" or "DED absent"). In the control arm, 18/82 completed the diabetic eye exam within 6 months, so the primary care gap closure rate was 22% (95%CI: 14%, 32%).

The difference of 78% (95% CI: 69%, 87%) in gap closure between control and intervention groups was statistically significant ($p < 0.001$). There were no statistically significant differences in care-gap closure rates by race, ethnicity, SES, or education. There were 3 participants in the control arm who did not return for clinic visits, were not able to be reached by phone or EHR-based messaging, and thus their eye exams were considered incomplete. Sensitivity analysis was performed excluding these 3 participants and also assuming they completed their eye exams, and the results were unchanged. [Supplementary Note 8; Tables S2 and S3].

## Secondary outcomes

In the intervention arm, 25 participants received a "DED present" output, and received the referral intervention; of these, 16/25 completed an ECP visit within 6 months, so the follow-through completion rate was 64% (95%CI: 43%, 81%). In the control arm, 18 participants visited the ECP, for a follow-through completion rate of 22% (95%CI: 14%, 32%) and none had DED. The difference of 42% (95%CI: 21%, 63%) in follow-through completion rates between control and intervention

**Table 3 | ACCESS primary outcome—completion of the diabetic eye exam (DED) (n = 163[a])**

| | AI group n = 81 | Control Group n = 82 | |
| --- | --- | --- | --- |
| | Completed DED screening | Completed DED screening | p-value[b] |
| N | 81 (100%) | 18 (22%) | <0.0001 |
| Age, mean (SD) | 15.3 (2.8) | 15.9 (2.9) | 0.20 |
| **Race** | | | 0.66 |
| Asian | 6 (7.4%) | 1 (5.6%) | |
| NH Black | 26 (32.1%) | 10 (55.6%) | |
| Hispanic | 5 (6.2%) | 1 (5.6%) | |
| NH White | 44 (54.3%) | 6 (33.3%) | |
| Male sex | 32 (39.5%) | 6 (33.3%) | 0.31 |
| **Household income** | | | 0.78 |
| $25,000 or less | 11 (13.6%) | 5 (27.8%) | |
| $25,000–$49,999 | 15 (18.5%) | 2 (11.1%) | |
| $50,000–$74,999 | 13 (16.0%) | 3 (16.7%) | |
| $75,000–$99,999 | 11 (13.6%) | 1 (5.6%) | |
| More than $100,000 | 21 (25.9%) | 6 (33.3%) | |
| Choose not to answer/ refused | 10 (12.3%) | 1 (5.6%) | |
| **Highest education** | | | 0.91 |
| Less than 12 years of high school | 2 (2.5%) | 1 (5.6%) | |
| High school/GED | 30 (37.0%) | 7 (38.9%) | |
| Associate's degree | 9 (11.1%) | 1 (5.6%) | |
| Undergraduate degree | 12 (14.8%) | 3 (16.7%) | |
| Post-graduate degree | 24 (29.6%) | 5 (27.8%) | |
| Unknown | 4 (4.9%) | 1 (5.6%) | |
| **Medicaid insurance** | 37 (45.7%) | 8 (44.4%) | 0.68 |
| Type 1 diabetes | 59 (72.8%) | 12 (66.7%) | 0.57 |
| Type 2 diabetes | 22 (27.2%) | 6 (33.3%) | 0.57 |
| Duration of diabetes (years), median (IQR) | 5.3 (3.4, 7.9) | 5.8 (2.4, 9.9) | 0.91 |
| HbA1c value at study visit, mean (SD) | 8.7 (2.3) | 8.2 (2.1) | 0.54 |
| Continuous glucose monitor use | 62 (76.5%) | 13 (72.2%) | 0.70 |

[a]Excludes one participant who was assigned to the control arm but inadvertently enrolled in another eye screening study

[b]p-values were calculated using Chi-Squared tests for categorical variables, Wilcoxon rank-sum test for duration of diabetes, and Student's t-tests for all other continuous variables. All statistical tests are two-sided.

groups was significant (p < 0.001). Further analysis comparing the 16 participants who completed follow-up after "DED present" to those that did not, showed that those who did not complete follow-up were more likely to be NH Black and have Medicaid insurance, but this was a small subgroup and not statistically significant.

A retina specialist review of the AI images (a level 4 reference standard) demonstrated an estimated sensitivity of 100% and specificity of 78.9%[33].

Participants reported a high level of satisfaction with autonomous AI, 92.5% were satisfied with the length of time it took to complete the exam, and 96% were satisfied with the experience. Of those who were in the intervention arm, 85% reported they would choose to do the AI-based eye exam in the future, and only 57%, an ECP-based diabetic eye exam (Table 4).

There were no adverse events during the study visits or fundus photography.

## Discussion

The results of the ACCESS trial confirmed our hypothesis that autonomous AI increases diabetic eye exam completion rates, and closes this care gap in a racially and ethnically diverse population of youth with diabetes, compared to standard of care. This result held true despite augmenting the standard of care referral in the control arm with deliberate education for the patient and caregiver regarding the importance of diabetic eye exams. Additionally, the results indicate that autonomous AI improves the likelihood of receiving follow-through eye care for those patients identified as having DED at the point-of-care compared to control. Care gap closure rates for the diabetic eye exam are an important component of value-based care through MIPS and HEDIS quality metrics, and the results show that autonomous AI can contribute to meeting these historically hard-to-achieve metrics, especially in racial/ethnic minority and under-resourced youth[34,35]. To our knowledge and based on a PubMed search [Supplementary Note 4], the present study is the first RCT to evaluate the role of autonomous AI in closing a guideline-based care gap.

Autonomous AI allows real-time, point-of-care diagnosis, whether in the primary care, endocrine, or other outpatient setting, can be integrated into the diabetes care workflow, and was included in the ADA standards of care for DED screening in adults as of 2020[36]. While teleretinal screening programs have improved care-gap closure rates over the last two decades[19,20,22,23,37], care-gap closure in adults nationwide remains at only 15.3% in 2018[7], despite all these efforts. In designing this randomized control trial, we considered the alternative comparator of teleretinal screening but chose the ECP exam as it is considered the standard of care, and both ECP and the AI system have been validated against patient outcome (as required by FDA)[33] and prognostic standards (Early Treatment of Diabetic Retinopathy Study (ETDRS and DRCR)), while teleretinal networks have not been validated against prognostic standards. Additionally, studies have shown large variability between telemedicine reading networks[38–40], and the sustainability and scalability of telemedicine have been limited. The advantages of autonomous AI are its point-of-care procedure, immediate results, and its scalability as it does not require additional clinical experts, ophthalmic oversight, or highly skilled operators. However, future studies should compare autonomous AI to teleretinal screening programs.

Compared to adults, the prevalence of DED is low in children. Although screening for DED is recommended, the reported DED care-gap closure in pediatrics ranges from 35-70%[10], and at the start of ACCESS, 79% of participants reported an eye exam at some time in the past. However, our previous studies demonstrated substantial racial/ethnic disparity, with non-white youth significantly less likely to have had a prior diabetic eye exam yet more likely to have DED[17]. The baseline data in this study confirmed these racial, ethnic, and socioeconomic gap closure disparities, similar to other studies demonstrating the wide range of disparities associated with social determinants of health in diabetic retinopathy screening[9,41]. However, the 100% care-gap closure rate by autonomous AI in this racially and ethnically diverse pediatric population could have the potential to reduce health disparities for vision loss from diabetes, furthering health equity, and deserves further study[42,43].

The high satisfaction and acceptance rates for autonomous AI in ACCESS, suggest that this racially and socioeconomically diverse patient population is comfortable with a "computer" or autonomous AI diagnosing their disease. Importantly, the use of AI did not introduce health disparities into care-gap closure.

While this study is an important step towards increased DED screening in youth, to prevent vision loss, patients identified with DED need follow-through care to manage and treat the disease[44]. Follow-through completion rates remain as low as 5-30% after tele-ophthalmology screening with referable DED[21,45,46], and educational

**Table 4 | Survey results**

| Survey question | AI (n = 80ᵃ) | Control with completed DED screening (n = 15ᵃ) | p-valueᵇ |
|---|---|---|---|
| My eyes are healthy (agree or strongly agree) | 82.5% | 73.3% | 0.47 |
| I know diabetes could have an impact on my eyesight (agree or strongly agree) | 91.1% | 93.3% | 0.78 |
| Having a diabetic eye exam regularly is important (agree or strongly agree) | 92.5% | 100.0% | 0.59 |
| How satisfied were you with the length of time it took to complete the diabetic eye exam? (very satisfied or satisfied) | 92.5% | 100.0% | 0.59 |
| How satisfied were you with the length of time it took to receive the results of your diabetic eye exam? (very satisfied or satisfied) | 95.0% | 93.3% | 0.59 |
| How satisfied were you that you received an easy to understand explanation of procedures before the eye exam? (very satisfied or satisfied) | 96.2% | 93.3% | 0.51 |
| How satisfied were you with the overall experience of having a diabetic eye exam done in the diabetes clinic? (very satisfied or satisfied) | 96.2% | n/a | n/a |
| For your next diabetic eye exam, how likely are you to choose a dilated eye exam at an eye care provider? (very likely or likely) | 57.0% | 80.0% | 0.15 |
| For your next diabetic eye exam, how likely are you to choose a point-of-care diabetic retinopathy screening exam using artificial intelligence? (very likely or likely) | 84.8% | 93.3% | 0.69 |

ᵃOne AI patient did not complete the survey; One AI patient completed only the first 5/9 questions; 4 SOC patients did not complete the survey.
ᵇFisher's exact test. All statistical tests are two-sided.

interventions have been shown to help in some cases[47]. In this study, referral augmented with targeted education was insufficient, as shown by the 22% follow-through at 6 months in the control arm. However, in the intervention arm, follow-through was significantly higher at 64%, as participants with "DED present" results were immediately informed of the diagnosis and provided education. Other studies have also demonstrated that point-of-care diagnosis and immediate results may affect patient behavior in seeking follow-through care[48,49]. An RCT of assistive AI that provided point-of-care results demonstrated a 30% increase in follow-through compared to telemedicine screening with deferred results in 3–5 days[49]. Another study reported improvement in follow-up rates from 35% to 72% with point-of-care diagnosis of referable DED[48].

Strengths of the present ACCESS study are its rigorous hypothesis-testing, pre-registered RCT design, as well as the scripted clinical education given to participants above the standard of care in the control arm, to maximize the likelihood of care gap closure in the control arm. The study sample was an adequate representation of youth with diabetes in the U.S. with respect to racial, ethnic, SES, education, and sex distribution, increasing the external validity of our results. Additionally, while this particular autonomous AI has been shown not to have racial bias, it is important to consider pigmentation of the retina that could introduce bias in an AI system for the diabetic eye exam[43].

Limitations of the present study are that the autonomous AI used is not FDA-cleared for use in ages 21 and under, though we showed in a previous study that the risk of false negatives is low[30]. Some of the participants in this study were familiar with autonomous AI diabetic eye exams from our prior study and may have been more accepting of participation in the current study[30]. Additionally, although pharmacologic dilation is not necessary for the pediatric population to obtain sufficient fundus imaging, this may not be applicable to adults where real-world studies have demonstrated higher rates of insufficient images without pharmacologic dilation[21,50].

In summary, the ACCESS trial shows that autonomous AI diabetic eye exams close more care gaps in youth with diabetes than the standard of care, and the availability of point-of-care AI diabetic eye exams may mitigate known screening disparities in racial/ethnic minority and under-resourced youth. Furthermore, sharing a diagnosis of referable disease at the point-of-care was associated with a higher rate of follow-through with eye care providers for management and treatment, potentially improving visual outcomes in this vulnerable population and advancing health equity in youth with diabetes.

## Methods
### Trial design
ACCESS is a hypothesis-driven, pre-registered, prospective parallel, RCT with a 1:1 allocation ratio that was conducted at the Johns Hopkins Pediatric Diabetes Center at two sites (Johns Hopkins Hospital and Mount Washington Pediatric Hospital) in Baltimore, Maryland, which serves a racially and ethnically diverse population. Participants were enrolled from November 24, 2021, through June 6, 2022, and follow-up was completed by Dec 6, 2022. The CONSORT requirements for RCT were followed[51]. ACCESS was pre-registered on ClinicalTrials.gov (NCT05131451). The study was approved by the Johns Hopkins IRB, the tenets of the Declaration of Helsinki were followed, and an independent Data Safety and Monitoring Board was established.

### Participants
Youth with T1D (11-21 years) or T2D (8-21 years) were eligible for inclusion if they met the criteria for DED screening per American Diabetes Association (ADA) 2021 guidelines[16], had no known DED, and had not had a diabetic eye exam within the last 6 months. Patients with maturity-onset diabetes of the young, cystic-fibrosis-related diabetes, known DED, or other pre-existing eye conditions (retinal disease, cataracts) were excluded from this study.

### Interventions
Potential study participants were approached in the diabetes clinic to confirm eligibility and then recruited by a study coordinator with written informed consent. Consented participants were randomized to either the control arm or the intervention arm. At the time of enrollment, the study coordinator collected baseline data, as well as 3 phone numbers from the participant in order to facilitate follow-up.

**Control arm: standard of care augmented with an educational intervention.** In the control arm, participants were referred to an eye care provider (ECP: optometrist or ophthalmologist) through a deliberate educational process by the study coordinator, in the form of a scripted educational intervention, including a paper handout guide on how to get a diabetic eye exam [Supplementary Note 1]. The goal of this intervention was to minimize the effect of the most commonly reported barrier, i.e., communication and confusion around the necessity of the diabetic eye exam[17]. Diabetic eye exam completion was achieved with ECP eye exam documentation. If the participant could not be reached to determine completion (despite calls/voicemails to all 3 phone numbers and EHR-based secure messaging) or if the exam had not been completed by 6 months, it was considered not completed.

**Intervention arm: autonomous AI**. In the intervention arm participants underwent the 5–10 min autonomous AI system diabetic eye exam without pharmacologic dilation[24]. The autonomous AI system (IDx-DR, Digital Diagnostics, Coralville, Iowa, USA) for diagnosing diabetic eye disease (DED) was US FDA De Novo authorized ("FDA approval") in 2018 for adults with diabetes[15]. The system diagnoses specific levels of diabetic retinopathy and diabetic macular edema (Early Treatment of Diabetic Retinopathy Study level 35 and higher, clinically significant macular edema, and or center-involved macular edema)[32,33], referred to as "referable DED"[34], that requires further management or treatment by an ophthalmologist or retina specialist. If the ETDRS level is 20 or lower and no macular edema is present, appropriate management is to retest in 12 months[35]. With this autonomous AI system, a medical diagnosis is made independently by the system without human oversight.

In this study, the participant's eyes were *not* pharmacologically dilated, as pilot studies found that pharmacologic dilation is unnecessary in youth[30]. The autonomous AI system guided the operator to acquire two color fundus images determined to be of adequate quality using an image quality algorithm[36], one each centered on the fovea and the optic nerve and guided the operator to retake any images of insufficient quality. This process requires approximately 10 min, after which the autonomous AI system reports one of the following within 60 s: "DED present, refer to a specialist", "DED not present, test again in 12 months", or "insufficient image quality". The latter response occurs when the operator is unable to obtain images of adequate quality after 3 attempts.

If the autonomous AI output was "DED absent," participants were informed the diabetic eye exam was normal, while if "DED present," (ETDRS level 35 or higher, and/or clinically significant, and or/center-involved macular edema) they received a deliberate educational process by the study coordinator in the form of a scripted educational intervention for follow-up eye care [Supplementary Note 2]. In either case, the diabetic eye exam was considered complete. If the AI output was "insufficient quality" the participant was referred for eye care.

The IDx-DR autonomous AI is not labeled for youth <22 years, as currently no autonomous AI for DED has been cleared for a pediatric population. To ensure safety and that no cases of disease would be missed, all images were also overread by a board-certified retina specialist.

**Follow-up procedures**. Participants who completed the diabetic eye exam (intervention arm: after autonomous AI exam; control arm: after documented completion of eye exam) were asked to fill out a survey on acceptability and satisfaction with the screening method. [Supplementary Note 3] Participants received a $25 gift card and parking pass for participation in the study.

### Randomization and masking
To prevent selection bias and ensure sample size balance between the groups and sites, stratified randomization (by site) was used, and participants were randomized in permutated block schedules of 4 and 6. Within each block, participants were randomized with a 1:1 allocation ratio to the control group and intervention group. This randomization sequence was created by a statistician unaffiliated with the study to ensure masking to the randomization scheme and was implemented by REDCAP's randomization software based on the participant's location[52,53]. After consent, the research coordinator entered the participant location and the randomization allocation was generated. All parties were masked to the allocation until the participant was randomized in the study, and then all parties were unmasked.

### Outcomes
In order to test the primary hypothesis, the pre-specified primary outcome was defined as the proportion of participants who completed a documented diabetic eye exam in each arm ("primary care gap closure rate"). In the control arm, this is the proportion of patients who completed a documented diabetic eye exam with an ECP within 6 months of randomization; in the intervention arm, this is the proportion of participants who completed the autonomous AI exam at the study visit.

The pre-specified secondary outcome ("follow-through completion rate") was defined as the proportion of participants who completed follow-through at the ECP in each arm. In the control arm, this was the proportion who completed the diabetic eye exam at ECP after referral, and in the intervention arm, the proportion who completed follow-up at the ECP after a "DED-present." This proportion assumes that control-arm patients who arrive at ECP for screening remain at ECP for management or treatment when found to have DED. Both outcomes were stratified by race, ethnicity, SES, and education level, using univariate and multivariate analysis to determine any differential effect on these categories. There were no changes in trial design or outcome after trial commencement.

**Data collection**. Data were collected from the electronic health record, specifically age, date of birth, sex at birth, race, ethnicity, type of diabetes, date of diabetes diagnosis, medication use (insulin, metformin, GLP1 agonist, etc.), form of insulin administration, use of continuous glucose monitor (CGM) and CGM data, blood pressure, height, weight, body mass index (BMI), presence of other diabetes-related complications (hyperlipidemia, hypertension, microalbuminuria), abnormal thyroid function, past four hemoglobin A1C readings (if available), diabetic eye exam history, medical history, family medical history, health insurance, and zip code. Parental education status and household income were self-reported by participants using a paper/pencil form.

### Sample size calculation
We assumed that a 20% difference in DED screening completion rates (care gap closure) would be clinically relevant. Based on our prior study[30], where baseline screening rates before AI were 49%, we assumed that with the educational intervention, screening rates for usual care would be closer to 60% in this study, and demonstrating a difference of 20% would be clinically relevant for AI screening. We calculated that a sample size of 164 ($n = 82$, $n = 82$ AI) would provide 80% power with a 2-tailed type-1 error of 0.05. Since randomization and study visits occurred at the same time there was little risk of attrition and thus the sample size was not expanded to account for attrition.

### Data governance
Although this was a low-risk clinical trial, an independent Data Safety and Monitoring Board was established to protect the interests of study participants and to preserve the integrity and credibility of the study data, based on pre-specified aims, thereby reducing any concerns that interim data could influence or bias the study results and interpretation. At the time of the DSMB meeting on 9/16/2022, all participants had already been enrolled in the trial, and the DSMB determined that there were no safety concerns and the study should continue to completion.

**Statistical analysis**. The primary outcome of care-gap closure between the randomization groups, and the secondary outcome of follow-through completion rate between the randomization groups were assessed by Pearson's chi-squared tests. Characteristics of the completed vs non-completed participants in each arm were assessed by Pearson's chi-squared tests, Wilcoxon rank-sum tests, and two sample *t*-tests, depending on the nature and distribution of each characteristic. A mixed multilevel multivariable logistic regression analysis, using the site as a nesting level to account for the clustering of

observations, was performed in order to examine the relationship between demographic characteristics and the odds of having a previous diabetic eye exam amongst the entire study cohort, adjusting for known covariates associated with DED and site. All analyses were performed using Stata 15.1 (StataCorp, College Station, TX).

## Reporting summary

Further information on research design is available in the Nature Portfolio Reporting Summary linked to this article.

## Data availability

Data from this study will be shared with bona fide researchers submitting a research proposal approved by the primary investigator. Data will be shared in a de-identified/anonymized format. The controlled access is implemented to safeguard the confidentiality and sensitive nature of the data. For access requests, please contact Risa Wolf at rwolf@jhu.edu. Access to the data will be granted upon review of the request, and researchers can expect a response within 90 days from the date of submission. Access to the data is contingent upon the completion of a data use agreement (DUA). The DUA outlines the terms and conditions for data usage, including restrictions on data sharing, publication, and any other pertinent considerations. Researchers granted access will be required to abide by the terms specified in the DUA to ensure the responsible and ethical use of the data.

## Code availability

The autonomous AI system (LumineticsCore, formally IDx-DR) used in this RCT is a deterministic (locked) medical device regulated and supervised by the US FDA under its De Novo regulation, and is commercially available from the manufacturer Digital Diagnostics or its distributors. The diagnostic algorithms that form the core of the medical device have been described in earlier publications. This system integrates many components, each of which contributes to its performance, safety, and usability by clinic staff without ML expertise in the real-world, as validated in the FDA pivotal trial.

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

## Acknowledgements

We thank the patients and their parents or guardians at the Johns Hopkins Pediatric Diabetes Center and Mount Washington Pediatric Hospital for their participation. This clinical trial was supported by the National Eye Institute of the National Institutes of Health under Award Number R01EY033233 and the Diabetes Research Connection to RMW. RC receives research support in part from an Unrestricted Grant from Research to Prevent Blindness, Inc. to the UW-Madison Department of Ophthalmology and Visual Sciences, and by the National Eye Institute of the National Institutes of Health under Award Number 5K23EY030911-03. The content is solely the responsibility of the authors and does not necessarily represent the official views of the funding agencies. Digital Diagnostics provided autonomous AI interpretations in-kind; no other support was received from Digital Diagnostics.

## Author contributions

Conception or design of the work: R.M.W., R.C., H.L., T.Y.A.L., L.P., M.D.A. Data collection: A.Z., L.B., A.A., D.P., E.B., T.Y.A.L., R.M.W., M.D.A. did not have access to the underlying data at any time. Data analysis and interpretation: E.B., L.P., R.M.W., H.L., R.C., M.D.A. Drafting the article: R.M.W., A.Z., D.P., E.B., L.P., M.D.A., H.L., R.C. Critical revision of the article: all final approval of the version to be published: All.

## Competing interests

M.D.A. reports the following conflicts of interest: Investor, Director, Consultant, Digital Diagnostics Inc., Coralville, Iowa, USA; patents and patent applications assigned to the University of Iowa and Digital Diag-nostics that are relevant to the subject matter of this manuscript; Chair Healthcare AI Coalition, Washington DC; member, American Academy of Ophthalmology (AAO) AI Committee; member, AI Workgroup Digital Medicine Payment Advisory Group (DMPAG); member, Collaborative Community for Ophthalmic Imaging (CCOI), Washington DC; Chair, Foundational Principles of AI CCOI Workgroup. R.M.W. reports receiving research support from Boehringer Ingelheim and Novo Nordisk outside the submitted work. The remaining authors declare no competing interests.
