## [Peer Review File · Nature Communications]

Autonomous artificial intelligence increases screening and follow-up for diabetic retinopathy in youth: the ACCESS randomized control trialREVIEWER COMMENTS

Reviewer #1 (Remarks to the Author):

The revised title of the paper is much better and more accurate than the previous version.

However, I still disagree with some of the conclusions and discussion as they are currently worded, as I do not believe that the data supports all of these conclusions.

1. With regard to the revised conclusion that "In summary, the ACCESS trial shows that autonomous AI diabetic eye exams close more care gaps in youth with diabetes than the standard of care, and may help mitigate screening disparities in minority and under-resourced youth", the authors suggest that Table 2 provides data in support of this statement. However, while Table 2 does indeed show that there were significant differences between participants who had and had not completed a previous DED screening at baseline, and that these differences had disappeared by the end of the trial, this actually refers to all 163 participants of the trial, who had either undergone autonomous AI screening, or standard of care ECP referral. Therefore, this data cannot be used to say that autonomous AI screening specifically closes more care gaps than standard of care. The table actually shows that the care gaps were closed by study participation, and by both the AI and control interventions.

2. The analysis of the socioeconomic characteristics of the 64% who followed through after AI screening and the 36% that did not follow through does show that there were no statistically significant differences between the two groups. However, there is a clear trend that patients who did not follow through were more likely to be NH Black, of lower household income, had lower education levels, and more likely to be on Medicaid insurance, than patients who did follow through. The lack of statistical significance here is clearly due to the small sample size of each small group. On the basis of these numbers I find it hard to justify the statement that "autonomous AI diabetic eye exams close more care gaps".

3. I appreciate the authors' discussion about the choice of comparator control arm, and their rationale for choosing an ECP exam as the control when designing the trial. I do still think that a telemedicine screening program would have been a fairer comparator for an AI point-of-care test. Regardless, I think that this discussion about the comparator arm needs to be included in the limitations and discussion of the manuscript.

4. Thank you for pointing out that the SEE study has previously reported the diagnostic accuracy, sensitivity and specificity of the autonomous AI system in the pediatric population. Nevertheless, even if these were not the primary outcomes in this RCT, I still think that they should be reported somewhere in the paper.

Reviewer #2 (Remarks to the Author):

Wolf and all have reasonably addressed all of my prior comments and suggestions and I believe this is an improved, solid paper of a well-done study.

My last suggestion from a health equity perspective is to add in 1-2 sentences either in the introduction or discussion section describing the limitations of using race and ethnicity as a variable in healthcare gaps or disparities. The authors, I believe, are not insinuating that the disparities in screening for DED by race/ethnicity are based on the differences in the melanin of the skin or genetics of the individuals. There needs to be a statement that the authors understand that race/ethnicity is a social construct, and their use of race/ethnicity throughout the paper is a surrogate for a complex, interrelated set of circumstances and lived experiences (e.g. type of neighborhood, parental stability,

social capital, extent of networks, minority stress, experiences of racism, etc.) that tend to be more common among those who are racial and ethnic minorities in this country.

Reviewer #3 (Remarks to the Author):

I would like to thank the authors for their thoughtful responses to my comments. I believe the introduction is substantially improved and most of my questions are answered satisfactorily.

The only issue that remains for me concerns the health disparities care-gap closure data in Table 2/3. I see that most other reviewers expressed concerns about this, and neither the responses nor manuscript revisions appear to address all of these concerns in my opinion. In particular from my perspective, the interpretation of Table 2 is difficult. I can see the loss of statistical significance for the race variables on the right side of Table 2, but the right side appears to lump together AI and control arms so the effect of the AI intervention is not visible. Am I missing the point of the table? Perhaps the headings could be improved. Table 3 seems to be a better place to draw conclusions regarding health disparities and care gaps between the two arms.

Reviewer #4 (Remarks to the Author):

The revised manuscript is much improved and satisfactorily addressed comments from previous reviewers. Here only a few minor comments are provided for improving the manuscript.

1. Line 139-140, it stated that "Diabetic eye exam completion" is defined based on the ECP eye exam document OR participant/caregiver report of eye exam completion. This is different from the primary outcome definition in Line 195 where it stated that "Diabetic eye exam completion" is purely based on the documented diabetic eye exam. This discrepancy should be solved.
2. Each participant has two eyes. Did the autonomous AI capture images both eyes simultaneously? Was the AI system report about the presence, absence of DED or image quality specific to each eye, or just report it at subject level?
3. Line 356, it states that false negative for DED using autonomous AI is very low from previous study. It will be good to report the false negative rate from this study too, since it seems data from image gradings by certified retina specialist are available.
4. In Line 357, it stated that some participants in this study were familiar with AI system from their participation in prior study. Could it be more explicit on how many participants exactly were familiar with AI system from prior study?
5. Tables 2 and 3, p-values were provided for comparison of each category of race, but not for the other variables with more than 2 levels including household income, parental education where a overall p-value was provided. I think providing overall p-value for race is sufficient to keep it consistent with other variables.

REVIEWER COMMENTS

Reviewer #1 (Remarks to the Author):

The revised title of the paper is much better and more accurate than the previous version.

However, I still disagree with some of the conclusions and discussion as they are currently worded, as I do not believe that the data supports all of these conclusions.

1. With regard to the revised conclusion that “In summary, the ACCESS trial shows that autonomous AI diabetic eye exams close more care gaps in youth with diabetes than the standard of care, and may help mitigate screening disparities in minority and under-resourced youth”, the authors suggest that Table 2 provides data in support of this statement. However, while Table 2 does indeed show that there were significant differences between participants who had and had not completed a previous DED screening at baseline, and that these differences had disappeared by the end of the trial, this actually refers to all 163 participants of the trial, who had either undergone autonomous AI screening, or standard of care ECP referral. Therefore, this data cannot be used to say that autonomous AI screening specifically closes more care gaps than standard of care. The table actually shows that the care gaps were closed by study participation, and by both the AI and control interventions.

Response: Thank you for this comment, and highlighting the need for clarification. The design of this RCT was *Intent to Screen*, and the outcome measure we are discussing was the fraction of patients in each arm who *completed the diabetic eye exam*. Given this design, the care gap could only be closed by completing the diabetic eye exam – either by AI, or going to the ECP (and not simply by study participation). To eliminate confusion, we have renamed Table 3 “*ACCESS primary outcome – completion of the diabetic eye exam*” to make sure the primary endpoint of this intent to screen RCT is clear, and that it clearly describes the primary outcome demonstrating 100% care gap closure in the autonomous AI group. We have updated Table 2 to only include the univariate analysis comparison of baseline participant characteristics in prior eye exam completion. Finally, we have added Table 4, which compares participant characteristics on baseline eye exam completion characteristics, compared to the entire cohort after completion of the ACCESS study. We believe this helps to clarify the conclusion that “*Autonomous AI diabetic eye exams close more care gaps in youth with diabetes than the standard of care,*” and “*mitigates screening disparities...*”

2. The analysis of the socioeconomic characteristics of the 64% who followed through after AI screening and the 36% that did not follow through does show that there were no statistically significant differences between the two groups. However, there is a clear trend that patients who did not follow through were more likely to be NH Black, of lower household income, had lower education levels, and more likely to be on Medicaid insurance, than patients who did follow through. The lack of statistical significant here is clearly due to the small sample size of each small group. On the basis of these numbers I find it hard to justify the statement that “autonomous AI diabetic eye exams close more care gaps”.

Response: Thank you for this important comment, we have included a more detailed discussion of this data in the Results:Secondary Outcomes section. Since more participants completed the diabetic eye exam in the autonomous AI arm compared to the usual care arm, the statement that “autonomous AI diabetic eye exams close more care gaps” is supported by the primary results.

“Further analysis comparing the 16 participants who completed follow-up after “DED present” to those that did not, showed that those who did not complete follow-up were more likely to be NH Black and have Medicaid insurance, but this was a small subgroup and not statistically significant.”

3. I appreciate the authors’ discussion about the choice of comparator control arm, and their rationale for choosing an ECP exam as the control when designing the trial. I do still think that a telemedicine screening program would have been a fairer comparator for an AI point-of-care test. Regardless, I think that this discussion about the comparator arm needs to be included in the limitations and discussion of the manuscript.

Response: Thanks for this suggestion, we have now included the discussion and rationale for the comparator arm in the discussion of the manuscript.

“ In designing this randomized control trial, we considered the alternative comparator of teleretinal screening, but chose the ECP exam as it is considered the standard of care, and both ECP and the AI system have been validated against patient outcome (as required by FDA)³⁸ and prognostic standards (Early Treatment of Diabetic Retinopathy Study (ETDRS and DRCR)), while teleretinal networks have not been validated against prognostic standards. Additionally, studies have shown large variability between telemedicine reading networks,³⁹⁻⁴¹ and the sustainability and scalability of telemedicine has been limited.”

References added:

Lin DY, Blumenkranz MS, Brothers RJ, Grosvenor DM. The sensitivity and specificity of single-field nonmydriatic monochromatic digital fundus photography with remote image interpretation for diabetic retinopathy screening: a comparison with ophthalmoscopy and standardized mydriatic color photography. *Am J Ophthalmol.* 2002;134(2):204-13.

Bhargava M, Cheung CY, Sabanayagam C, et al. Accuracy of diabetic retinopathy screening by trained non-physician graders using non-mydratic fundus camera. *Singapore Med J.* 2012;53(11):715-719.

Zhou P, Eltemsah L, Bahrainian M, Prichett L, Liu TYA, Wolf RM, Channa R. Assessment of Trained Image Grader Performance in Screening for Retinopathy Among Youth With Diabetes. *J Diabetes Sci Technol.* 2022 Nov;16(6):1580-1581. doi: 10.1177/19322968221120240. Epub 2022 Sep 1. PMID: 36047654; PMCID: PMC9631538.

4. Thank you for pointing out that the SEE study has previously reported the diagnostic accuracy, sensitivity and specificity of the autonomous AI system in the pediatric population. Nevertheless, even if these were not the primary outcomes in this RCT, I still think that they

should be reported somewhere in the paper.

Response: We appreciate the reviewers concern for patient safety. We do not report the sensitivity and specificity here as it was not a primary endpoint of this trial, and other studies have assessed this in a more rigorous manner with consensus grading of three retinal specialists. In this trial, since the IDx-DR (now LumineticsCore) system is not yet FDA approved for youth <22years of age, the retinal specialist overread was to ensure safety and that no cases of disease would be missed, and there were no false negative results (no cases missed) and no safety issues with the study as reviewed by the Data Safety Monitoring Board (DSMB). We are currently conducting a larger safety and efficacy trial for LumineticsCore for youth, funded by NIH and JDRF (clinicaltrials.gov: NCT05463289) where the AI will be validated against the prognostic standard as required by FDA,³⁸ and we will publish the results in the near future. We have modified the methods accordingly.

Methods: *“To ensure safety and that no cases of disease would be missed, all images were also overread by a board certified retina specialist.”*

Reviewer #2 (Remarks to the Author):

Wolf and all have reasonably addressed all of my prior comments and suggestions and I believe this is an improved, solid paper of a well-done study.

My last suggestion from a health equity perspective is to add in 1-2 sentences either in the introduction or discussion section describing the limitations of using race and ethnicity as a variable in healthcare gaps or disparities. The authors, I believe, are not insinuating that the disparities in screening for DED by race/ethnicity are based on the differences in the melanin of the skin or genetics of the individuals.

Response: Thank you for this important point. While the issue is much broader, in this specific case of the retina exam, there has been and is indeed concern about the melanin content of the retina (pigment epithelium) - associated with skin melanin content - affecting the accuracy of the AI for the diabetic eye exam differentially. Such AI bias (caused by differences in retinal pigment) has the potential to increase rather than decrease health disparities. While this

particular autonomous AI was shown not to have such bias based on skin pigment in adults, it is important to consider. This is highlighted in the introduction:

*“...showing its safety, efficacy **and lack of racial and ethnic bias** for diagnosing DED in adults with diabetes with 87% sensitivity and 91% specificity. ²⁴”*

The following reference has been added to the discussion: Abramoff, MD, Tarver, E. M, Loyo-Barrios, N, Trujillo, S, Char, D. Considerations for Addressing Bias in Artificial Intelligence for Health Equity, Nat Dig Med 2023 (in press).

There needs to be a statement that the authors understand that race/ethnicity is a social construct, and their use of race/ethnicity throughout the paper is a surrogate for a complex, interrelated set of circumstances and lived experiences (e.g. type of neighborhood, parental stability, social capital, extent of networks, minority stress, experiences of racism, etc.) that tend to be more common among those who are racial and ethnic minorities in this country.

Response: Thank you for this helpful comment. We have added in an additional sentence to the discussion that highlights that in addition to retinal melanin, the wider issue of social determinants of health in contributing to disparities in diabetic retinopathy screening and reference a recent comprehensive review on this topic.

“The baseline data in this study confirmed these racial, ethnic and socioeconomic gap closure disparities, similar to other studies demonstrating the wide range of disparities associated with social determinants of health in diabetic retinopathy screening.”

Reference added: Patel D, Ananthakrishnan A, Lin T, Channa R, Liu TYA, Wolf RM. Social Determinants of Health and Impact on Screening, Prevalence, and Management of Diabetic Retinopathy in Adults: A Narrative Review. J Clin Med. 2022 Nov 30;11(23):7120. doi: 10.3390/jcm11237120. PMID: 36498694; PMCID: PMC9739502.

Reviewer #3 (Remarks to the Author):

I would like to thank the authors for their thoughtful responses to my comments. I believe the introduction is substantially improved and most of my questions are answered satisfactorily.

The only issue that remains for me concerns the health disparities care-gap closure data in Table 2/3. I see that most other reviewers expressed concerns about this, and neither the responses nor manuscript revisions appear to address all of these concerns in my opinion. In particular from my perspective, the interpretation of Table 2 is difficult. I can see the loss of statistical significance for the race variables on the right side of Table 2, but the right side appears to lump together AI and control arms so the effect of the AI intervention is not visible. Am I missing the point of the table? Perhaps the headings could be improved. Table 3 seems to be a better place to draw conclusions regarding health disparities and care gaps between the two arms.

Response: Thank you for your helpful comments on this topic. We have updated and clarified the title of Table 2 to *“Table 2: ACCESS Patient Characteristics by Previous Eye Exam, and Overall Completion of the Diabetic Eye Exam in this Trial.”* Table 2 is important to show the

baseline demographics of those with and without a prior diabetic eye exam. However, we have removed all discussion regarding the right side of Table 2.

Reviewer #4 (Remarks to the Author):

The revised manuscript is much improved and satisfactorily addressed comments from previous reviewers. Here only a few minor comments are provided for improving the manuscript.

1. Line 139-140, it stated that “Diabetic eye exam completion” is defined based on the ECP eye exam document OR participant/caregiver report of eye exam completion. This is different from the primary outcome definition in Line 195 where it stated that “Diabetic eye exam completion” is purely based on the documented diabetic eye exam. This discrepancy should be solved.

Response: Thanks for highlighting this discrepancy: All patient/caregiver reports of the diabetic eye exam were confirmed with documentation, so we have clarified this in the manuscript.

2. Each participant has two eyes. Did the autonomous AI capture images both eyes simultaneously? Was the AI system report about the presence, absence of DED or image quality specific to each eye, or just report it at subject level?

Response: The autonomous AI system captures the eye images consecutively, and while the AI makes a DED interpretation per eye, as per the FDA labelling for adults, reports results at the subject level.

3. Line 356, it states that false negative for DED using autonomous AI is very low from previous study. It will be good to report the false negative rate from this study too, since it seems data from image gradings by certified retina specialist are available.

Response: There were no false negative results (no cases missed) and therefore no safety issues reported to the Data Safety Monitoring Board (DSMB) for this trial. At the request of FDA we are currently conducting a larger safety and efficacy trial for LumineticsCore for youth, funded by NIH and JDRF (clinicaltrials.gov: NCT05463289) where the AI will be validated against the prognostic standard as required by FDA,³⁸ and will publish the results in the near future.

4. In Line 357, it stated that some participants in this study were familiar with AI system from their participation in prior study. Could it be more explicit on how many participants exactly were familiar with AI system from prior study?

Response: There were 53 patients who participated in the SEE study in 2018-2019, that were eligible to participate and enrolled in the ACCESS trial in 2021-2022.

5. Tables 2 and 3, p-values were provided for comparison of each category of race, but not for the other variables with more than 2 levels including household income, parental education where a overall p-value was provided. I think providing overall p-value for race is sufficient to keep it consistent with other variables.

Response: We have updated Table 3 to only include an overall p-value for race. Given that there are significant differences by race/ethnicity in Table 2, we opted to leave the p-values for race.

REVIEWER COMMENTS

Reviewer #1 (Remarks to the Author):

Thank you for the clarifications and changes made. I appreciate the significant changes that have been made to the paper in revision, and it is a good manuscript describing an important study, that is much improved. However, respectfully, I still disagree with two points, which in my opinion remain inadequately addressed.

1. The wording of the article from the first version has been significantly improved, but there are still multiple statements in the article that suggest that the results support that autonomous AI closes care gaps particularly in under-served or minority youth, which I think is misleading based on the results.

For example:

- "and the results show that autonomous AI can contribute to meeting these historically hard to achieve metrics, especially in racial/ethnic minority and under-resourced youth"
- "However, the 100% care-gap closure rate by autonomous AI entirely removed these disparities, confirming earlier pilot results. Thus, autonomous AI may reduce health disparities for visual loss from diabetes in youth, furthering health equity"
- "Importantly, the racial, ethnic, and SES disparities in care gap closure rates present at baseline, were no longer present at the end of the trial."
- "In summary, the ACCESS trial shows that autonomous AI diabetic eye exams close more care gaps in youth with diabetes than the standard of care, and may help mitigate screening disparities in minority and under-resourced youth"

Yet, the results simply show that the autonomous AI closes more of the care gap than standard of care in youth with diabetes. I do not believe the results support any particularly greater benefit in under-served or minority youth. As the authors state in the results, "There were no statistically significant differences in care-gap closure rates by race, ethnicity, SES or education." The changes to the tables do not address this either. It seems to me that the new Table 2 is simply the first half of the old Table 2, and the new Table 4 is the old Table 2 reproduced in entirety. I understand the distinction in the standard of care arm of the trial between study participation (receiving an ECP referral and education) and completing the diabetic eye exam (going to the ECP). However, that is not the issue here.

As an analogy, this is like comparing two drugs in an RCT for glycemic control, where the primary outcome measure shows that drug 1 is superior to drug 2 in improving glycemic control. Then showing a table of the baseline characteristics of the whole study cohort (including both treatment arms), and showing that there are differences in baseline socioeconomic characteristics of the study cohort when stratified by glycemic control. Then, showing that after study completion, these differences in baseline characteristics of the whole study cohort (including both treatment arms) when stratified by glycemic control are no longer present. And then, using this to support the statement that drug 1 is better than drug 2 at improving glycemic control in patients with disadvantaged socioeconomic characteristics. That is not accurate. Unless the effectiveness of drug 1 is compared against drug 2 specifically in a disadvantage socioeconomic group, I do not think that such claims can be made.

I would argue personally that the primary results of the study clearly support the conclusion that the autonomous AI intervention results in a higher diabetic eye exam completion rate than the standard of care in youth with diabetes, and avoid making conclusions in relation to the relative effectiveness of AI versus the standard of care in minority or under-served youth or other socioeconomic characteristics, unless this aspect can be strongly justified.

2. I also understand that the diagnostic accuracy, sensitivity and specificity of the autonomous AI system were not the primary outcomes in this RCT. However, I still think that they should be reported,

either as a secondary outcome or in supplementary. I see that another reviewer has raised the same point, and I do not think that the fact that this is being studied in another, separate, trial means that the results within this trial should not be reported.

Reviewer #2 (Remarks to the Author):

The only comment would be to include some of the explanation in their response below in the actual manuscript to better explain the disparities due to actual melanin/pigmentation- for readers who are not as familiar with disparities innate in AI for retinal exams.

The authors response:

Thank you for this important point. While the issue is much broader, in this specific case of the retina exam, there has been and is indeed concern about the melanin content of the retina (pigment epithelium) - associated with skin melanin content - affecting the accuracy of the AI for the diabetic eye exam differentially. Such AI bias (caused by differences in retinal pigment) has the potential to increase rather than decrease health disparities. While this particular autonomous AI was shown not to have such bias based on skin pigment in adults, it is important to consider.

Reviewer #3 (Remarks to the Author):

I would like to thank the authors for addressing my comments and revising Tables 2-4. These changes, in addition to the responses to Reviewer 1, have fully addressed my concerns. I have no further requests for clarification or revision.

Reviewer #4 (Remarks to the Author):

The authors satisfactorily addressed all previous comments. The manuscript is improved.

Reviewer #1 (Remarks to the Author):

Thank you for the clarifications and changes made. I appreciate the significant changes that have been made to the paper in revision, and it is a good manuscript describing an important study, that is much improved. However, respectfully, I still disagree with two points, which in my opinion remain inadequately addressed.

1. The wording of the article from the first version has been significantly improved, but there are still multiple statements in the article that suggest that the results support that autonomous AI closes care gaps particularly in under-served or minority youth, which I think is misleading based on the results.

For example:

- “and the results show that autonomous AI can contribute to meeting these historically hard to achieve metrics, especially in racial/ethnic minority and under-resourced youth”
- “However, the 100% care-gap closure rate by autonomous AI entirely removed these disparities, confirming earlier pilot results. Thus, autonomous AI may reduce health disparities for visual loss from diabetes in youth, furthering health equity”
- “Importantly, the racial, ethnic, and SES disparities in care gap closure rates present at baseline, were no longer present at the end of the trial.”
- “In summary, the ACCESS trial shows that autonomous AI diabetic eye exams close more care gaps in youth with diabetes than the standard of care, and may help mitigate screening disparities in minority and under-resourced youth”

Yet, the results simply show that the autonomous AI closes more of the care gap than standard of care in youth with diabetes. I do not believe the results support any particularly greater benefit in under-served or minority youth. As the authors state in the results, “There were no statistically significant differences in care-gap closure rates by race, ethnicity, SES or education.” The changes to the tables do not address this either. It seems to me that the new Table 2 is simply the first half of the old Table 2, and the new Table 4 is the old Table 2 reproduced in entirety. I understand the distinction in the standard of care arm of the trial between study participation (receiving an ECP referral and education) and completing the diabetic eye exam (going to the ECP). However, that is not the issue here.

As an analogy, this is like comparing two drugs in an RCT for glycemic control, where the primary outcome measure shows that drug 1 is superior to drug 2 in improving glycemic control. Then showing a table of the baseline characteristics of the whole study cohort (including both treatment arms), and showing that there are differences in baseline socioeconomic characteristics of the study cohort when stratified by glycemic control. Then, showing that after study completion, these differences in baseline characteristics of the whole study cohort (including both treatment arms) when stratified by glycemic control are no longer present. And then, using this to support the statement that drug 1 is better than drug 2 at improving glycemic control in patients with disadvantaged socioeconomic

characteristics. That is not accurate. Unless the effectiveness of drug 1 is compared against drug 2 specifically in a disadvantage socioeconomic group, I do not think that such claims can be made.

I would argue personally that the primary results of the study clearly support the conclusion that the autonomous AI intervention results in a higher diabetic eye exam completion rate than the standard of care in youth with diabetes, and avoid making conclusions in relation to the relative effectiveness of AI versus the standard of care in minority or under-served youth or other socioeconomic characteristics, unless this aspect can be strongly justified.

Response: Thank you for the insightful comments and explanation. The example provided made a lot of sense, and made us realize that we had not clarified that we purposely conducted this study in Baltimore, Maryland, as this is a minority majority city, and the cohort of pediatric diabetes patients is more racially and ethnically diverse than in most practices, in order to study the effects of autonomous AI in these populations. As shown in Table 1, >40% of the participants were black or Hispanic. The background for this is that we previously found in the SEE study that non-white youth were less likely to undergo diabetic eye exams yet more likely to have diabetic retinopathy, which is a primary reason for doing this study in this setting and with this diverse population. To this point, we have updated the manuscript to clarify this in the introduction and methods, and have also removed or amend the statements the reviewer referred to above. Additionally the paragraph about the exploratory analysis performed has been removed, in addition to Table 4. Please see below for edits made to the manuscript based on the comments above.

- “and the results show that autonomous AI can contribute to meeting these historically hard to achieve metrics, especially in racial/ethnic minority and under-resourced youth” - This statement remains now that we clarified the population this study was conducted in.

- “However, the 100% care-gap closure rate by autonomous AI ***in this racially and ethnically diverse pediatric population could have the potential to*** ~~entirely removed these disparities, confirming earlier pilot results. Thus, autonomous AI may~~ reduce health disparities for vision loss from diabetes, furthering health equity, ***and deserves further study.***”

- “Importantly, the racial, ethnic, and SES disparities in care gap closure rates present at baseline, were no longer present at the end of the trial.” - **Removed**

- “In summary, the ACCESS trial shows that autonomous AI diabetic eye exams close more care gaps in youth with diabetes than the standard of care, ***and availability of point of care AI diabetic eye exams may mitigate known*** screening disparities in minority and under-resourced youth”

Removed Table 4 and associated results.

2. I also understand that the diagnostic accuracy, sensitivity and specificity of the

autonomous AI system were not the primary outcomes in this RCT. However, I still think that they should be reported, either as a secondary outcome or in supplementary. I see that another reviewer has raised the same point, and I do not think that the fact that this is being studied in another, separate, trial means that the results within this trial should not be reported.

Response: We have included the diagnostic accuracy of the AI system for this sample of 81 participants compared to the retina specialist review. Sensitivity was 100% and specificity was 78.9%. However, we note that diagnostic accuracy, sensitivity and specificity of the autonomous AI system were not a primary outcome of this RCT, nor was the study powered to assess this given the small sample size of 81 participants, and thus we would recommend relying on the previously published diagnostic accuracy in the SEE study. We completely agree that the sensitivity and specificity of the autonomous AI system are important, but to address this key question, the gold standard, level 1 prognostic standard with images interpreted at a central reading center should be performed. This requires a large amount of resources and funding, which is currently underway in a larger safety and efficacy trial for LumineticsCore for youth, funded by NIH and JDRF (clinicaltrials.gov: NCT05463289) where the AI will be validated against the prognostic standard as required by FDA,³⁸ and we will publish the results in the near future.

Results: *“Retina specialist review of the AI images (a level 4 reference standard) demonstrated an estimated sensitivity of 100% and specificity of 78.9%.”* Reference: Abramoff MD, Cunningham B, Patel B, Eydelman MB, Leng T, Sakamoto T, Blodi B, Grenon SM, Wolf RM, Manrai AK, Ko JM, Chiang MF, Char D; Collaborative Community on Ophthalmic Imaging Executive Committee and Foundational Principles of Ophthalmic Imaging and Algorithmic Interpretation Working Group. Foundational Considerations for Artificial Intelligence Using Ophthalmic Images. *Ophthalmology*. 2022 Feb;129(2):e14-e32. doi: 10.1016/j.ophtha.2021.08.023. Epub 2021 Aug 31. PMID: 34478784; PMCID: PMC9175066.

Reviewer #2 (Remarks to the Author):

The only comment would be to include some of the explanation in their response below in the actual manuscript to better explain the disparities due to actual melanin/pigmentation- for readers who are not as familiar with disparities innate in AI for retinal exams.

The authors response:

Thank you for this important point. While the issue is much broader, in this specific case of the retina exam, there has been and is indeed concern about the melanin content of the retina (pigment epithelium) - associated with skin melanin content - affecting the accuracy of the AI for the diabetic eye exam differentially. Such AI bias (caused by differences in retinal pigment) has the potential to increase rather than

decrease health disparities. While this particular autonomous AI was shown not to have such bias based on skin pigment in adults, it is an important aspect to consider when evaluating the performance of AI.

Response: Thank you, we have added the following statement to the strengths section of the discussion

“Additionally, this particular autonomous AI was shown not to have racial bias, but it is important to consider pigmentation of the retina that could introduce bias in an AI system for the diabetic eye exam.” (Reference: Abramoff MD, Tarver ME, Loyo-Berrios N, Trujillo S, Char D, Obermeyer Z, Eydelman MB; Foundational Principles of Ophthalmic Imaging and Algorithmic Interpretation Working Group of the Collaborative Community for Ophthalmic Imaging Foundation, Washington, D.C.; Maisel WH. Considerations for addressing bias in artificial intelligence for health equity. NPJ Digit Med. 2023 Sep 12;6(1):170. doi: 10.1038/s41746-023-00913-9. PMID: 37700029)

Reviewer #3 (Remarks to the Author):

I would like to thank the authors for addressing my comments and revising Tables 2-4. These changes, in addition to the responses to Reviewer 1, have fully addressed my concerns. I have no further requests for clarification or revision.

Thank you

Reviewer #4 (Remarks to the Author):

The authors satisfactorily addressed all previous comments. The manuscript is improved.

Thank you

REVIEWERS' COMMENTS

Reviewer #1 (Remarks to the Author):

Thank you for the clarifications. I am satisfied that my concerns have been addressed.